# L-Arginine and Cardioactive Arginine Derivatives as Substrates and Inhibitors of Human and Mouse NaCT/Nact

**DOI:** 10.3390/metabo12040273

**Published:** 2022-03-22

**Authors:** Daniela B. Surrer, Martin F. Fromm, Renke Maas, Jörg König

**Affiliations:** Institute of Experimental and Clinical Pharmacology and Toxicology, Friedrich-Alexander-Universität Erlangen-Nürnberg, Fahrstrasse 17, 91054 Erlangen, Germany; daniela.surrer@fau.de (D.B.S.); martin.fromm@fau.de (M.F.F.); renke.maas@fau.de (R.M.)

**Keywords:** uptake, arginine derivatives, L-arginine, cardioactive arginine metabolites, transport inhibition

## Abstract

The uptake transporter NaCT (gene symbol *SLC13A5*) is expressed in liver and brain and important for energy metabolism and brain development. Substrates include tricarboxylic acid cycle intermediates, e.g., citrate and succinate. To gain insights into the substrate spectrum of NaCT, we tested whether arginine and the cardioactive L-arginine metabolites asymmetric dimethylarginine (ADMA) and L-homoarginine are also transported by human and mouse NaCT/Nact. Using HEK293 cells overexpressing human or mouse NaCT/Nact we characterized these substances as substrates. Furthermore, inhibition studies were performed using the arginine derivative symmetric dimethylarginine (SDMA), the NaCT transport inhibitor BI01383298, and the prototypic substrate citrate. Arginine and the derivatives ADMA and L-homoarginine were identified as substrates of human and mouse NaCT. Transport of arginine and derivatives mediated by human and mouse NaCT were dose-dependently inhibited by SDMA. Whereas BI01383298 inhibited only human NaCT-mediated citrate uptake, it inhibits the uptake of arginine and derivatives mediated by both human NaCT and mouse Nact. In contrast, the prototypic substrate citrate inhibited the transport of arginine and derivatives mediated only by human NaCT. These results demonstrate a so far unknown link between NaCT/Nact and L-arginine and its cardiovascular important derivatives.

## 1. Introduction

Transport proteins are important for the uptake, distribution, and excretion of endogenous substances and xenobiotics. Generally, they can be divided into two major groups: uptake transporters mostly belonging to the superfamily of SLC (solute carrier) transporters [1] and export pumps, belonging to the superfamily of ABC (ATP-binding cassette) transporters [2,3]. One important family for the transport of endogenous metabolism intermediates is the SLC family SLC13, the human sodium-sulfate/carboxylate cotransporter family comprised of five different members (SLC13A1–SLC13A5 [4]).

The human sodium-coupled citrate transporter NaCT (gene symbol *SLC13A5*), also known as human INDY (INDY = I am not dead yet) is an electrogenic uptake transporter highly expressed in liver and brain [5]. NaCT mediates the sodium-dependent uptake of tricarboxylic acid cycle (TCA) intermediates such as citrate and succinate [6] from blood into hepatocytes or neurons. In Caenorhabditis elegans and Drosophila melanogaster, reduced expression of NaCT promotes longevity due to caloric restriction [7,8,9]. Studies with Nact (-/-) knock out mice demonstrated that deletion of this uptake transporter mimics aspects of dietary restriction and protects these mice against adiposity and insulin resistance [10]. Furthermore, the absence of this transporter led to defective bone and tooth development in mice [11].

Besides its important role for energy homoeostasis, NaCT also seems to be important for brain function and development because several mutations in the *SLC13A5* gene encoding NaCT are associated with epileptic encephalopathies [11]. Epileptic encephalopathies refer to a clinically and genetically heterogeneous group of disorders characterized by abnormal electro-encephalograms, psychomotor delay, and seizures. Several mutations in the *SLC13A5* gene have been characterized regarding their functional consequences [12,13,14] and for some of them, it has been demonstrated that these mutations alter the localization of the [14].

Interestingly, despite 75% amino acid identity and the capability of transporting citrate, Nact (-/-) knockout mice do not show the epileptic phenotype seen in humans with the loss of function mutations in the *SLC13A5* gene [15]. Furthermore, human and mouse NaCT/Nact also differ regarding their affinity to the prototypic substrate citrate. Whereas a relatively high affinity constant (K_m_ value) for human NaCT-mediated citrate transport of 2 254 µM [16] has been calculated, the K_m_ value for mouse Nact is much lower (49 µM [10]). In line with these differences between human NaCT and rodent Nact, recently a potent transport inhibitor (BI01383298) for human NaCT-mediated uptake has been characterized [17]. This inhibitor irreversibly inhibits human NaCT-mediated citrate transport but has no effect on mouse Nact-mediated transport. These findings demonstrate that human NaCT and rodent Nact differ regarding their transport mechanism. This may be interesting in terms of the consequences of human NaCT deficiency. Thus far, the effect of NaCT absence or transport inhibition regarding the role of this transporter for energy homoeostasis has been explained by reduced citrate uptake into liver [16]. In contrast, the effect for brain function and development can hardly be explained by an impaired citrate transport function into neurons because this reduced citrate uptake is comparable between the human genetic deficiency and the animal knock-out mouse models. This leads to the assumption that human and rodent NaCT/Nact may differ in their ability to transport other, yet unknown substrates.

One important group of endogenous substances related to liver and brain metabolism and therefore pending on transporters mediating the uptake from blood into cells is arginine and the arginine metabolites L-homoarginine and asymmetric dimethylarginine (ADMA). L-homoarginine is an endogenous, non-proteinogenic amino acid enhancing endothelial function [18], and lower L-homoarginine plasma concentrations are associated with cardiovascular and all-cause mortality [19]. In contrast, elevated plasma concentrations of the structurally related arginine derivative and uremic toxin ADMA have been linked to cardiovascular events and mortality [20]. Recently, we could demonstrate that the renal uptake transporter OATP4C1 (gene symbol *SLCO4C1*) is capable of transporting arginine and both arginine derivatives [21]. In addition, these substances have been characterized as substrates for several other transporters including the cationic amino acid transporters CAT1, CAT2A, and CAT2B (for an overview on transporter mediating arginine and arginine derivative transporter see: [22]). Some of these transporters are also expressed in liver and brain. Furthermore, the liver is important in the metabolism of arginine and arginine metabolites, e.g., by expressing arginases [23,24] or by degrading the uremic toxin ADMA [25] by the enzyme dimethylarginine-dimethylaminohydrolase I (DDAH1), highly expressed in liver. Studies with patients suffering from hyperargininemia, a rare autosomal disease characterized by arginase deficiency have shown [24] that these patients have multiple clinical symptoms, including seizures and intellectual impairment [26,27]. These results point to an important role of arginine and arginine metabolites in the brain.

Therefore, the aim of this study was to investigate if arginine and the arginine derivatives L-homoarginine and ADMA are substrates of human NaCT and mouse Nact. For this, we used established and characterized HEK cells recombinantly overexpressing human NaCT [16] and mouse Nact [10] and characterized NaCT/Nact-mediated transport of arginine, ADMA and L-homoarginine. Furthermore, the inhibition of this NaCT-/Nact-mediated transport by the commercially available transport inhibitor BI01383298 [17], specific for human NaCT and by the prototypic substrate citrate, has been investigated.

## 2. Results

### 2.1. L-Arginine and Arginine Derivatives as Substrates of Human and Mouse NaCT/Nact

First, we tested the amino acid L-arginine as well as the cardioactive L-arginine derivatives asymmetric dimethylarginine (ADMA) and L-homoarginine as possible substrates. In established uptake assays using stably transfected HEK-NaCT, HEK-Nact, and HEK-Co cells, we investigated human NaCT- and mouse Nact-mediated uptake of arginine, ADMA and L-homoarginine. As demonstrated in Figure 1 both human NaCT and mouse Nact are capable of mediating the highly significant transport of all three substances. Highly significant uptake could be demonstrated for substrate concentrations ranging from 1 to 50 µM (Figure 1), which are in the range of the plasma concentrations of ADMA and L-homoarginine. Based on these initial data, we then calculated the kinetic constants of human (Figure 2) and mouse (Figure 3) NaCT-/Nact-mediated arginine (2A and 3A), ADMA (2B and 3B), and L-homoarginine (2C and 3C) transport.

### 2.2. Inhibition of NaCT/Nact-Mediated Transport of L-Arginine and Arginine Derivatives by Symmetric Dimethylarginine (SDMA)

In the first inhibition experiment we investigated, if the NaCT-/Nact-mediated uptake of arginine and both arginine metabolites can be inhibited by symmetric dimethylarginine (SDMA). Given that SDMA is structurally closely related to ADMA we used substrate concentrations of 50 µM, which are around the calculated K_m_ value for human NaCT-mediated transport. These results (Figure 4) demonstrate that NaCT from both species can dose-dependently be inhibited by this arginine derivative. 

### 2.3. Inhibition of NaCT-/Nact-Mediated Transport of L-Arginine and Arginine Derivatives by BI01383298

Next, we investigated whether the potent NaCT inhibitor BI01383298 also inhibits the transport of L-arginine and arginine derivatives, mediated by human NaCT and mouse Nact. This transporter has been described as being specific for inhibiting the human NaCT protein and not the rodent Nact transporter. These findings could be confirmed using citrate as substrate (Figure 5). Furthermore, transport of arginine, ADMA, and L-homoarginine mediated by human NaCT could also be inhibited (Figure 5A). In contrast to citrate as substrate, BI01383298 was also capable of significantly inhibiting the arginine and arginine derivatives transport mediated by mouse Nact (Figure 5B). 

### 2.4. Inhibition of NaCT/Nact-Mediated Transport of L-Arginine and Arginine Derivatives by the Prototypic Substrate Citrate

Finally, we investigated whether the transport of the newly identified substrates can be inhibited by the prototypic substrate citrate. Whereas human NaCT-mediated transport of arginine, ADMA, and L-homoarginine can be inhibited by citrate (Figure 6A), no transport inhibition for the same substrates could be detected for mouse Nact-mediated uptake (Figure 6B) even at the highest tested citrate concentration of 5000 µM. 

## 3. Discussion

In this manuscript we identified arginine, ADMA, and L-homoarginine as substrates for human and mouse NaCT/Nact but observed differential effects on transport inhibition. Arginine and L-homoarginine are both important for liver and brain function [28] and the liver is involved in the metabolism of the uremic toxin ADMA [25]. Therefore, we used established and well-characterized HEK293 transfectants recombinantly overexpressing human NaCT or mouse Nact [10,16] and investigated whether both transporters mediated the cellular uptake of the three substances (Figure 1). Based on net uptake values, the kinetic transport constants (K_m_ values) were calculated (Figure 2 and Figure 3) demonstrating that human NaCT and mouse Nact have a relatively high affinity for arginine and arginine derivatives compared to other transporters of these substances. In humans, L-arginine and arginine derivatives have been characterized as substrates for at least 18 transport proteins including members of the SLC families SLC3, SLC6, SLC7, SLC21/SLCO, SLC22, SLC25, SLC38, and SLC47 (for a detailed overview of transporters, mediating the uptake of L-arginine and L-arginine derivatives see: [22]). Interestingly, for arginine the calculated K_m_ values range from high affinity arginine transporters such as the SLC6 family member ATB^0,+^ (SLC6A14, [29]) with a calculated K_m_ value of 80 µM to the relatively low affinity of CAT2A (SLC7A2) with a K_m_ value of 3000 µM [30,31]. Furthermore, expression of arginine transporters was demonstrated in almost every tissue and cell investigated (e.g., CAT1; gene symbol *SLC7A1* seems to be ubiquitously expressed) explaining the relatively high background of arginine and arginine derivative uptake observed in the HEK-Co control transfectants (Figure 1). Interestingly, the K_m_ value for human NaCT-mediated arginine uptake of 99.8 µM is in the range of the arginine plasma concentration suggesting that this transport may also be important for arginine uptake into hepatocytes and neurons whereas for both arginine derivatives investigated, the calculated K_m_ values are above the calculated plasma concentrations. Moreover, NaCT appears to have a higher affinity towards ADMA and L-homoarginine than many other transporters listed as key transporters for these substances [22].

NaCT/Nact is expressed in human and mouse liver [5] localized to the basolateral hepatocyte membrane. In humans, the liver plays an important role not only for arginine metabolism (e.g., by arginases expressed in hepatocytes) but is also involved in the metabolism of ADMA [32] and the structurally related uremic toxin SDMA [33]. SDMA can inhibit NaCT-/Nact-mediated arginine, ADMA, and L-homoarginine uptake (Figure 4), suggesting that SDMA could also be a substrate of both transporters. Elevated plasma concentrations of L-arginine and the methylated derivatives have been reported in patients with impaired liver function [34,35], suggesting that the uptake of these substances is an important step in their hepatic handling. These data are supported by studies with patients suffering from severe alcoholic hepatitis [36]. In this study, the authors investigated 52 patients and found that plasma ADMA and SDMA levels were increased in these patients and that this results in higher portal blood pressures. These increased plasma levels may arise from decreased ADMA and SDMA breakdown, highlighting the role of uptake transporters for these substances in the basolateral membrane of hepatocytes. An association of low L-homoarginine and elevated methylarginines plasma concentrations with liver dysfunction and mortality has also been investigated in patients with chronic liver disease [35]. In this study the authors demonstrated that in cirrhotic patients low L-homoarginine plasma concentrations were inversely related to long-term mortality, while high ADMA and SDMA plasma concentrations were risk markers for long-term mortality. Interestingly, it has been demonstrated that in Nact (-/-) knock out mice arterial blood pressure and heart rate were reduced compared to wild type mice [37]. In this study, the authors claimed that this effect was due to novel mechanisms regulating catecholamine biosynthetic pathways and sympathoadrenal tone. A possible link between catecholamines and ADMA has been found in patients with rheumatoid arthritis. In these patients, a correlation between the urinary excretion of DHPG (3,4-dihydroxyphenylglycol, the major urinary metabolite of the catecholamine norepinephrine) and DMA (dimethylamine, the degradation product of ADMA) was found, suggesting a potential connection of catecholamine and ADMA metabolism [38]. 

Besides its expression in liver, NaCT/Nact is also expressed in brain. Several studies have demonstrated that the central nervous system and the brain are important targets for arginine and L-homoarginine. Especially studies with patients suffering from hyperargininemia have shown that arginine and L-homoarginine are important for proper brain function. Hyperargininemia is a rare autosomal-recessive hereditary disorder caused by a deficiency of the enzyme arginase, which hydrolyses arginine to ornithine and urea [24]. Compared to the normal range, the plasma concentrations of arginine, L-homoarginine, and other guanidine compounds are much higher in hyperargininemic patients [39]. Patients suffering from this disorder have multiple clinical symptoms including seizures, intellectual impairment, spasticity, and coma [26,27]. These severe neurological and cognitive problems could be confirmed in arginase-deficient mice [40] suggesting that the brain is an important target for arginine, L-homoarginine, and potentially other arginine derivatives. Interestingly, brain tissue levels of L-homoarginine were estimated to be three- to fivefold higher than levels measured in the cerebrospinal fluid, suggesting an effective uptake of this compound into neurons [26]. Furthermore, L-homoarginine levels differ between humans and rodents. Whereas in the human brain L-homoarginine concentrations of 1.52 ± 0.45 nmol/g tissue were measured [41], the amount in rat the brain was much lower (0.1–1.02 nmol/g tissue), suggesting that L-homoarginine has different effects on brain function of both species and perhaps explaining the differences observed between human patients lacking functional NaCT and Nact (-/-) knock out animals. 

Differences between human NaCT- and rodent Nact-mediated uptake could also be confirmed in this study. Whereas the arginine derivative and uremic toxin SDMA inhibited both human NaCT- and mouse Nact-mediated arginine, ADMA, and L-homoarginine uptake, citrate only inhibits the uptake of these substances mediated by human NaCT. Given that the K_m_ value for human NaCT-mediated citrate transport is relatively high (2 254 µM [16]), high citrate concentrations were also used for these inhibition experiments. Even if a significant transport inhibition by citrate for human NaCT-mediated arginine, ADMA, and L-homoarginine transport could be measured, this inhibition was not concentration-dependent (Figure 6A). However, despite the fact that the K_m_ value for mouse Nact-mediated citrate transport is magnitudes lower (49 µM [10]) compared to the human K_m_ value, no transport inhibition by citrate could be detected (Figure 6B) demonstrating a different effect of citrate on human NaCT- and mouse Nact-mediated transport. Furthermore, the effect of the transport inhibitor BI01383298 was also different on human NaCT- and mouse Nact-mediated transport. As expected, BI01383298 inhibited only human NaCT-mediated citrate uptake [17] and had no effect on mouse Nact-mediated citrate transport (Figure 5). In contrast, BI01383298 inhibited both human NaCT- and mouse Nact-mediated arginine, ADMA, and L-homoarginine transport (Figure 5) demonstrating marked functional differences in substrate recognition and transport between human NaCT and mouse Nact. This could explain differences between patients with loss of function mutations in the *SLC13A5* gene and Nact (-/-) knock-out mouse models. Whereas loss of function mutations in humans cause severe complications with encephalopathy, neonatal epilepsy, and teeth hypoplasia, mice lacking this transporter do not have this brain phenotype. Unfortunately, neither human NaCT nor mouse Nact has been crystallized, but based on modeling data compared to VcIndy [42], a related transporter expressed in bacteria, structural differences between human and mouse NaCT have been identified explaining the different transport and inhibitor properties of both transporters [15]. Recently, cryo-electron microscopy structures of human NaCT in complexes with citrate or two small molecule inhibitors (PF06649298 and PF06678419) were published [43]. Based on this structural analysis, the effect of mutations on NaCT-mediated transport could be explained. 

Taken together, we have identified arginine and both the cardioactive arginine metabolites asymmetric dimethylarginine (ADMA) and L-homoarginine as substrates for both human NaCT and mouse Nact. Furthermore, we could demonstrate that transport of these newly identified substrates could be modified by other substances with differential effects observed for human and mouse NaCT/Nact. These findings may explain the observed differences between human and mouse NaCT-mediated transport. Given the relatively high affinities of NaCT towards L-arginine and its derivatives as compared to other transport proteins so far assumed to be key transporters for these substances, the physiological role of NaCT-mediated arginine, ADMA, and L-homoarginine transport into liver and neurons deserves further elucidation. 

## 4. Materials and Methods

### 4.1. Materials

[^3^H]labeled ADMA (25 Ci/mmol) was from BIOTREND Chemikalien GmbH (Cologne, Germany), [^3^H]labeled L-homoarginine (6 Ci/mmol) was from ViTrax (St. Jefferson, MO, USA), [^3^H]labeled L-arginine (43 Ci/mmol) and [14C]citrate (113 mCi/mmol) were from American Radiolabeled Chemical (St. Louis, MO, USA). Unlabeled ADMA, SDMA, and L-arginine were from Enzo Life Sciences GmbH (Lörrach, Germany), unlabeled L-homoarginine was purchased from Acros Organics (Pittsburgh, PA, USA) and unlabeled citrate was obtained from Fisher Bioreagents (Waltham, MA, USA). The NaCT inhibitor BI 01383298 (1-(3,5-Dichlorophenylsulfonyl)-N-(4-fluorobenzyl)piperidine-4-carboxamide) was purchased from MedChem Express (Monmouth Junction, NJ, USA) Sodium butyrate was obtained from Merck KGaA (Darmstadt, Germany), Dulbecco’s phosphate buffered saline, fetal bovine serum, geneticin disulfate (G418), penicillin-streptomycin solution, 0.05%-trypsin-EDTA (0.02%) solution and minimum essential medium were from life Technologies GmbH (Darmstadt, Germany). All other substances were purchased from Sigma-Aldrich (St. Louis, MO, USA) with highest grade available. BCA Pierce Protein Assay Kit was obtained from Life Technologies and 12-well cell culture plates were from Greiner Bio-One (Frickenhausen, Germany).

### 4.2. Cell Culture

For this study HEK-NaCT (HEK293 cells stably expressing the human uptake transporter NaCT), HEK-Nact (HEK293 cells stably expressing the mouse uptake transporter Nact) and the control cell line HEK-Co (transfected with the empty expression vector and selected under the same conditions) were used. All cell lines were cultured in minimal essential medium, supplemented with 10% heat-inactivated fetal bovine serum, 100 U/mL penicillin and 100 µg/mL streptomycin. Geneticin (G418—800 µg/mL) was used as selection antibiotic. Cells were cultured at 37 °C and 5% CO_2_. Routinely, the cells were subcultured using trypsin 0.05%-EDTA (0.02%) solution.

### 4.3. Transport and Inhibition Studies

HEK-NaCT (expressing human NaCT–INDY), HEK-Nact (expressing mouse Nact–Indy) and HEK-Co cells were seeded into poly-D-lysine coated 12-well plates with an initial density of 5 × 10^5^ cells per well. After seeding, the cells were incubated at 37 °C and 5% CO_2_ before the protein expression was induced by adding medium supplemented with 10 mM sodium butyrate [44]. Before uptake experiments, cells were washed with prewarmed uptake buffer (5 mM KCl, 1 mM K_2_HPO_4_, 142 mM NaCl, 1.5 mM CaCl_2_, 1.2 mM MgSO_4_, 5 mM glucose, and 12.5 mM HEPES, pH 7.3). Radiolabeled substrates were dissolved in uptake buffer and unlabeled substances were added in the respective concentrations for the uptake and inhibition experiments. The initial uptake experiments using L-arginine, ADMA, or L-homoarginine were carried out using 1 µM, 10 µM, and 50 µM substrate concentrations. To determine the kinetic parameters (K_m_ values) of the cellular uptake of L-arginine, ADMA and L-homoarginine substrate concentrations between 1 and 500 µM were used. The cells were incubated with the uptake buffer for 10 min at 37 °C as described [10]. Then, the cells were washed three times with ice-cold uptake buffer to remove radioactivity bound to the cell membrane. Finally, the cells were lyzed with 0.2% SDS, the intracellular radioactivity was determined by liquid scintillation counting (TriCarb 2800, Perkin Elmer Life and Analytical Sciences GmbH, Rodgau, Germany), and the protein concentrations per well were determined using the bicinchoninic acid assay (BCA Protein Assay Kit, Thermo Fisher Scientific, Bonn, Germany).

To test the inhibitory effect of the arginine metabolite symmetric dimethylarginine (SDMA) on NaCT-/Nact-mediated L-arginine, ADMA, and L-homoarginine uptake, radiolabeled substances were used in a concentration of 50 µM and 100 µM and 500 µM SDMA was added into the uptake buffer. The NaCT inhibitor BI01383298 was tested in a concentration of 10 µM. Given that it has been described that BI01383298 irreversibly inhibited human NaCT-mediated transport, HEK-NaCT, HEK-Nact and HEK-Co cells were preincubated with the inhibitor for 30 min prior uptake experiments. Then, the inhibitor-containing uptake buffer was removed, the cells were washed twice with prewarmed uptake buffer, and a normal uptake assay using citrate (1 µM), L-arginine (10 µM), ADMA (10 µM) or L-homoarginine (10 µM) was performed. For inhibition studies with arginine, ADMA and L-homoarginine as substrate for human NaCT and mouse Nact, substrate concentrations of 1 µM were used and citrate (500 µM, 1000 µM, 2500 µM, and 5000 µM) was added as potential uptake inhibitor.

### 4.4. Statistical Analysis

All experiments were performed at least in two independent experimental setups with *n* = 3 each (*n* = 6). Transport data were normalized to the protein concentrations of the cell lysates. All data were expressed as means ± SEM. Statistical significance was considered to be given at a *p*-value < 0.05. Net uptake was calculated by subtracting the accumulation of the respective compound into HEK-Co cells from the accumulation into the transporter expressing cells (HEK-NaCT or HEK-Nact). No significant differences could be detected in the uptake rates into HEK-Co cells (e.g., in the setup with and without added inhibitor) demonstrating that net uptake values represent NaCT-/Nact-mediated transport. The percentagewise inhibition of NaCT- or Nact-mediated L-arginine, ADMA, L-homoarginine, or citrate transport was calculated by comparing the transport to a control experiment conducted without added inhibitor (net transport = 100%) and changes in transport rates were analyzed using one-way ANOVA with Bonferroni’s multiple comparison test.

Affinity constants (K_m_ values) and maximal transport velocity (V_max_ values) were calculated using Michaelis–Menten kinetics and GraphPad Prism (Version 5.01, 2007, GraphPad Software, San Diego, CA, USA).

## Figures and Tables

**Figure 1 metabolites-12-00273-f001:**
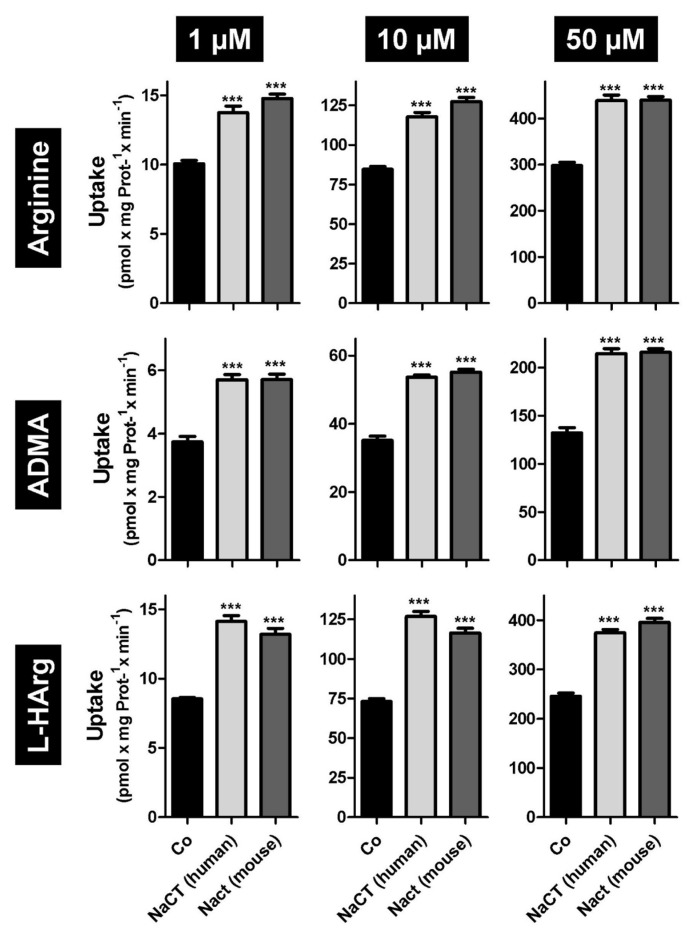
Initial uptake experiments with HEK-NaCT (NaCT human—light grey bars), HEK-Nact (Nact mouse—dark grey bars) and HEK-Co/418 (Co—black bars) cells using arginine, asymmetric dimethyl arginine (ADMA), and L-homoarginine (L-HArg) as substrate. All substances were added in three different concentrations (1 µM, 10 µM and 50 µM) *** *p* < 0.001 vs uptake in Co.

**Figure 2 metabolites-12-00273-f002:**
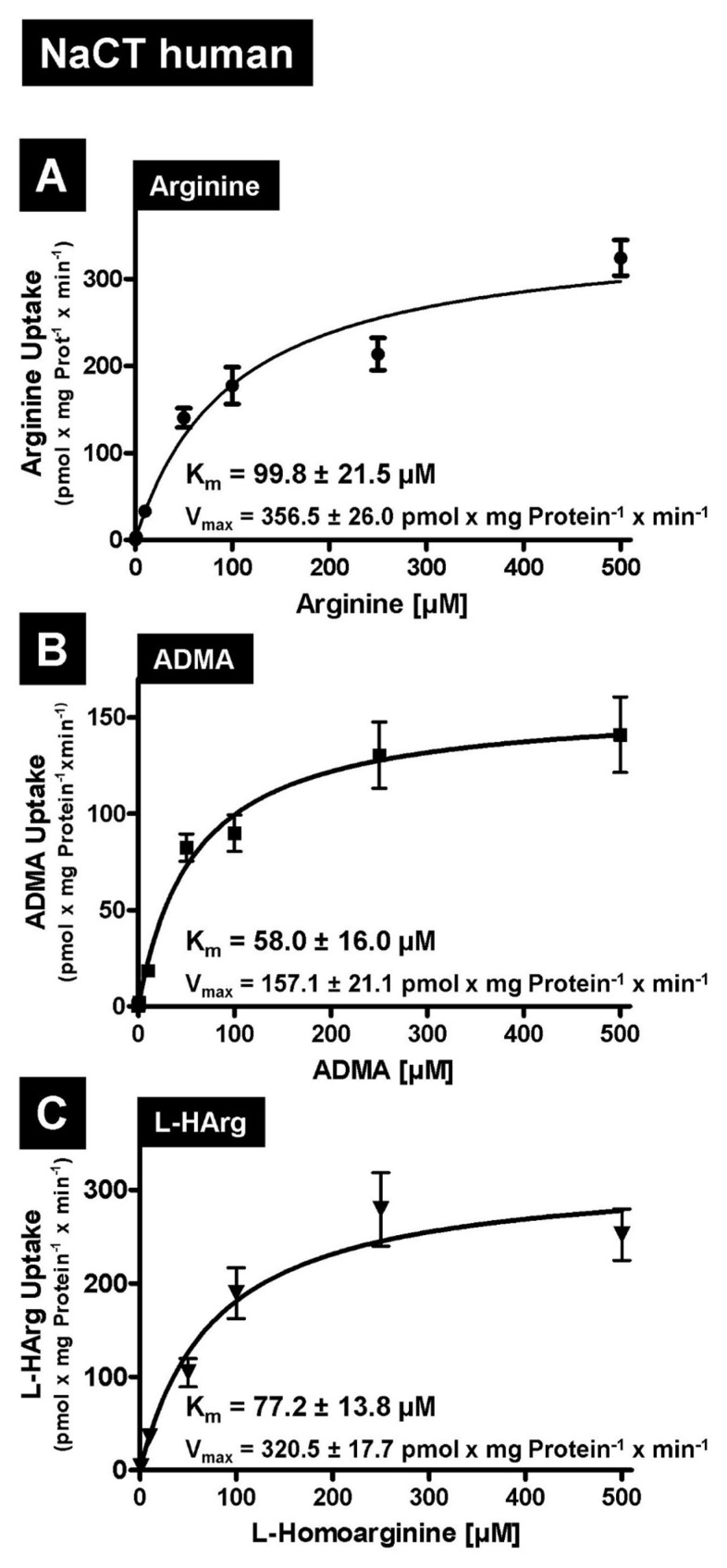
Determination of the kinetic transport constants (K_m_ values) and of the transport velocities (V_max_ values) of human NaCT-mediated Arginine (**A**), ADMA (**B**) and L-homoarginine (**C**) uptake. Uptake is based on net uptake values by subtracting the uptake into HEK-Co control cells from the uptake into HEK-NaCT cells.

**Figure 3 metabolites-12-00273-f003:**
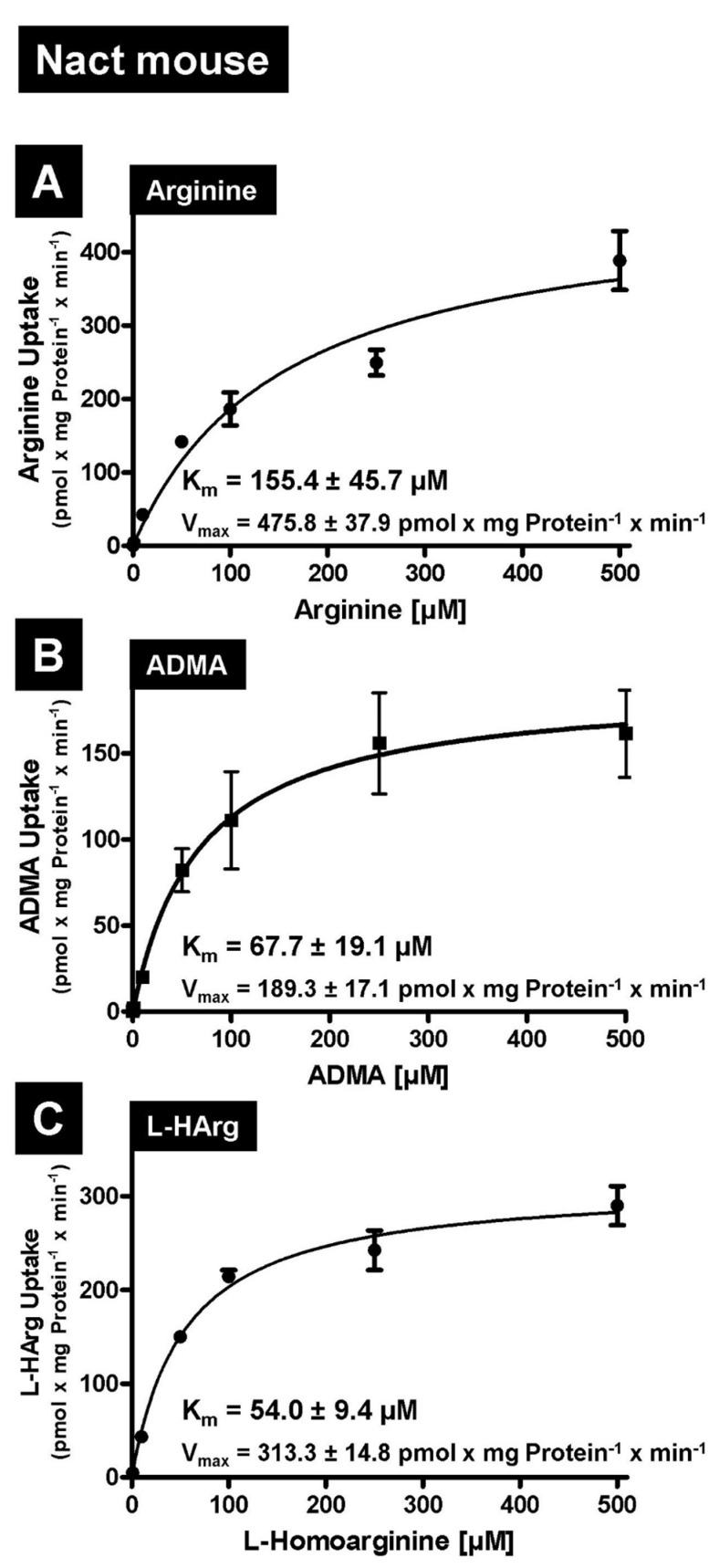
Determination of the kinetic transport constants (K_m_ values) and of the transport velocities (V_max_ values) of mouse Nact-mediated Arginine (**A**), ADMA (**B**) and L-homoarginine (**C**) uptake. Uptake is based on net uptake values by subtracting the uptake into HEK-Co control cells from the uptake into HEK-Nact cells.

**Figure 4 metabolites-12-00273-f004:**
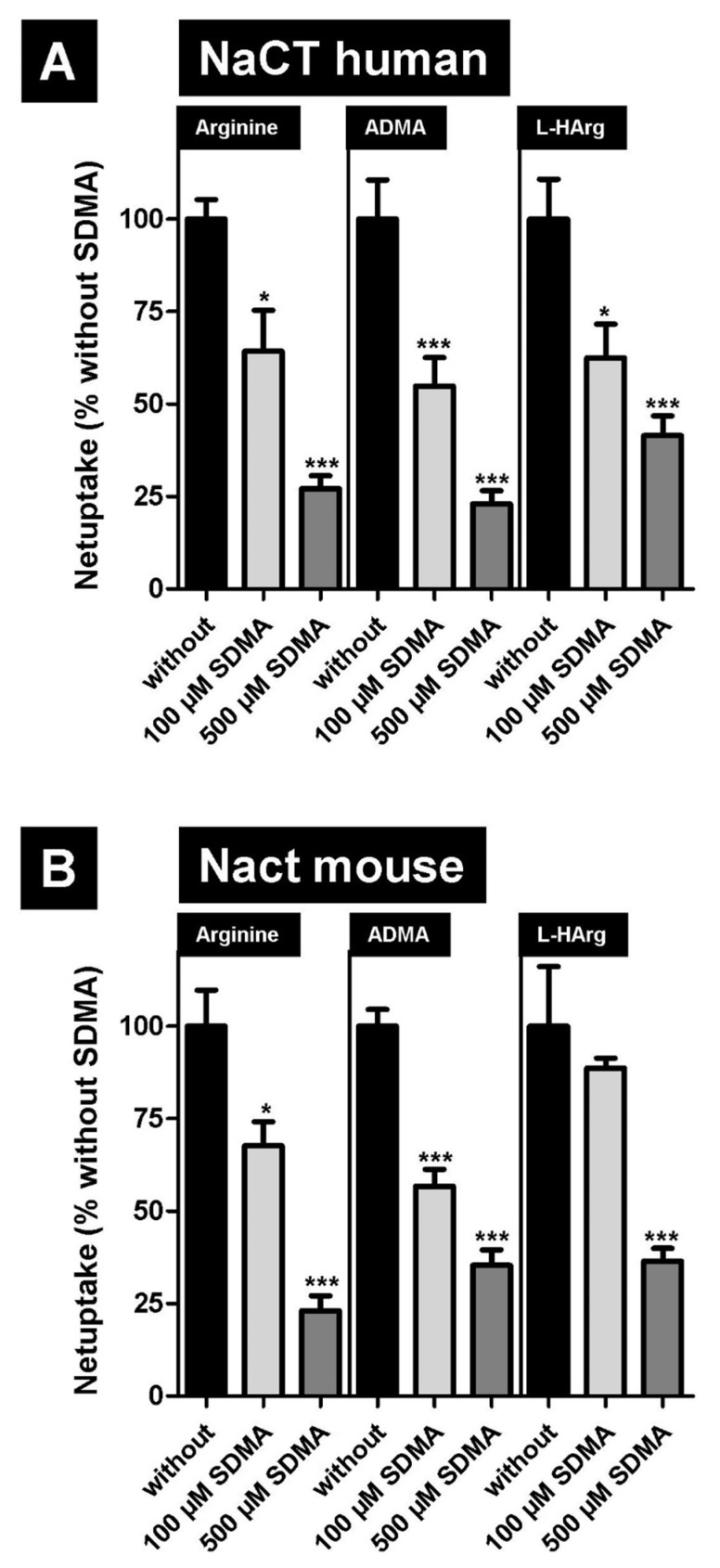
Inhibition of human (**A**) NaCT- and mouse (**B**) Nact-mediated arginine (50 µM), ADMA (50 µM), and L-homoarginine (50 µM) uptake by symmetric dimethylarginine (SDMA). SDMA was applied in two concentrations (100 µM—light grey bars and 500 µM—dark grey bars). Net uptake without added SDMA was set to 100% (black bars). * *p* > 0.01; *** *p* < 0.001 vs. uptake without SDMA.

**Figure 5 metabolites-12-00273-f005:**
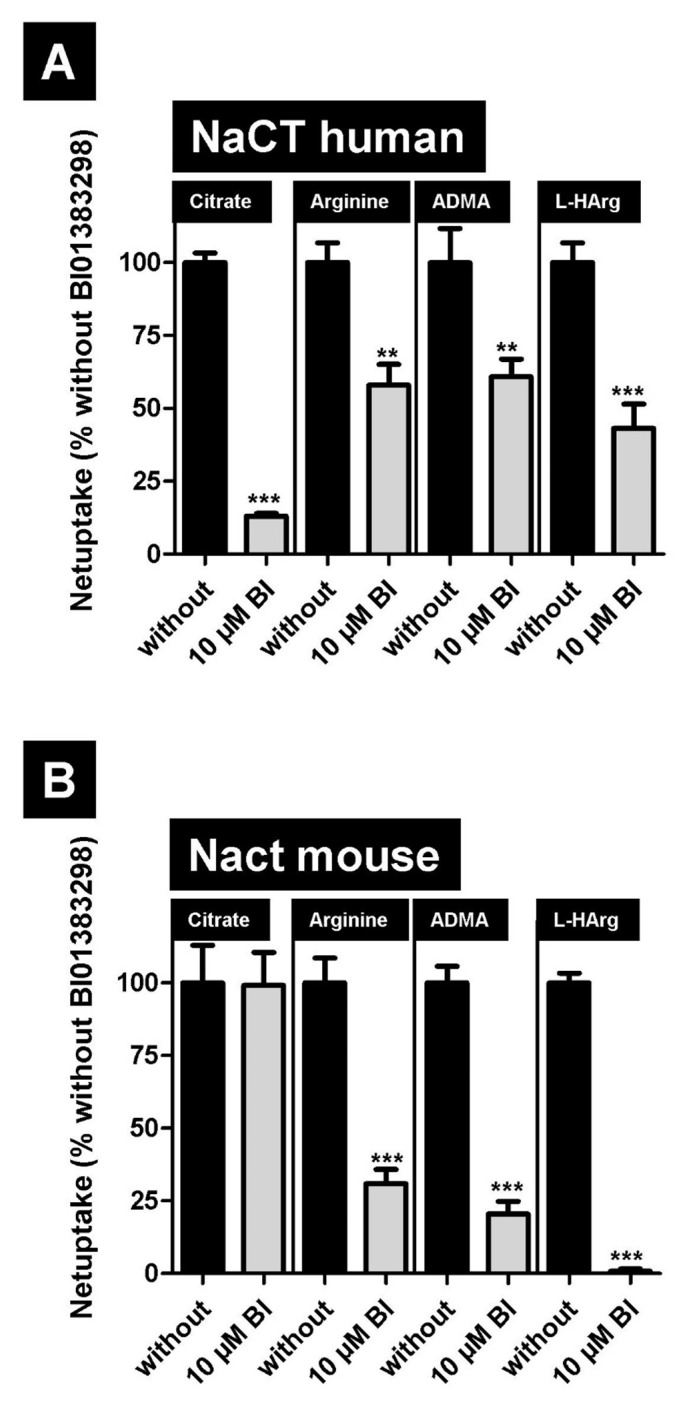
Inhibition of (**A**) human NaCT- and (**B**) mouse Nact-mediated citrate (1 µM), arginine (10 µM), asymmetric dimethylarginine (ADMA, 10 µM) and L-homoarginine (L-HArg, 10 µM) uptake by BI01383298. Cells were preincubated with 10 µM BI01383298 for 30 min before uptake experiments. Net uptake without added inhibitor was set to 100% (black bars). ** *p* < 0.005, *** *p* < 0.001 vs. uptake without BI01383298.

**Figure 6 metabolites-12-00273-f006:**
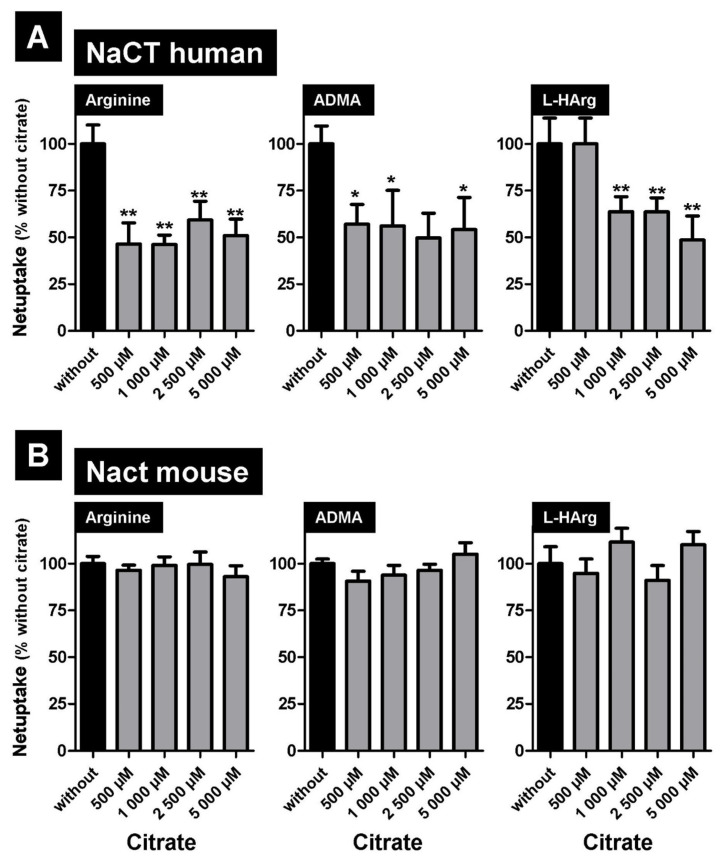
Inhibition of human (**A**) NaCT- and (**B**) mouse Nact-mediated arginine uptake of arginine (1 µM), asymmetric dimethylarginine (ADMA, 1 µM) and L-homoarginine (1 µM) by increasing concentrations of citrate. Net uptake without added citrate was set to 100% (black bars). * *p* < 0.01; ** *p* < 0.005 vs. uptake without added substance.

## Data Availability

All data is contained within this article.

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
