# Peer review of "L-Arginine and Cardioactive Arginine Derivatives as Substrates and Inhibitors of Human and Mouse NaCT/Nact"

_metabolites, 2022, doi:10.3390/metabo12040273_

Round 1

Reviewer 1 Report

Since the dose response is not clear in Fig. 5 and Fig. 6A, the authors should perform experiments with the lower and the higher doses of BI and the lower doses of Citrate.

Reviewer 2 Report

The authors have been demonstrated that SLC13A5, a sodium-dependent citrate transporter, has the ability to transport arginine and its derivatives. Furthermore, they found species differences in the substrate specificity of SLC13A5 in functional inhibition studies. However, due to the higher endogenous transport activities for arginine and its derivatives compared with netuptake by SLC13A5, it would require careful interpretation.

1) Is there a sodium dependence of the uptakes of arginine and its derivatives mediated by SLC13A5?

2) In Figure 5: It should be shown whether BI inhibited intrinsic uptakes of arginine and its derivatives in control HEK293 cells.

3) Figure 1: The uptake activities in the selected stable cell lines treated with sodium butyrate should be confirmed simply in HEK293 cells transiently expressed human and mouse SLC13A5, which makes the findings in this study more clear.

4) The transport activities of mouse and human SLC13A5 as citrate transporters in cells used in this study are not indicated in this manuscript.

Round 2

Reviewer 1 Report

Although the anthors did not reply to my concerns completely, I agree with their reply.

Reviewer 2 Report

I have no more comment.